# Psychological Distress among Undergraduate Dental Students in Saudi Arabia and Its Coping Strategies—A Systematic Review

**DOI:** 10.3390/healthcare9040429

**Published:** 2021-04-07

**Authors:** Sanjeev B. Khanagar, Ali Al-Ehaideb, Ahmed Jamleh, Khansa Ababneh, Prabhadevi C. Maganur, Satish Vishwanathaiah, Mohammed Adel Awawdeh, Sachin Naik, Abdulaziz A. Al-Kheraif, Shilpa Bhandi, Alessio Zanza, Luca Testarelli, Shankargouda Patil

**Affiliations:** 1Preventive Dental Science Department, College of Dentistry, King Saud Bin Abdulaziz, University for Health Sciences, Riyadh 11426, Saudi Arabia; sanjeev.khanagar76@gmail.com (S.B.K.); ehaideba@ksau-hs.edu.sa (A.A.-E.); ababnehk@ksau-hs.edu.sa (K.A.); m97a97@gmail.com (M.A.A.); 2King Abdullah International Medical Research Center, Riyadh 11426, Saudi Arabia; aojamleh@gmail.com; 3Dental Services, King Abdulaziz Medical City-Ministry of National Guard Health Affairs, Riyadh 11426, Saudi Arabia; 4Restorative and Prosthetic Dental Sciences Department, College of Dentistry, King Saud Bin Abdulaziz University for Health Sciences, Riyadh 11426, Saudi Arabia; 5Department of Preventive Dental Sciences, Division of Pediatric Dentistry, College of Dentistry, Jazan University, Jazan 44512, Saudi Arabia; prabhadevi.maganur@gmail.com (P.C.M.); drvsatish77@gmail.com (S.V.); 6Dental Biomaterials Research Chair, Dental Health Department, College of Applied Medical Sciences, King Saud University, Riyadh 11433, Saudi Arabia; sachinnaiksln@gmail.com (S.N.); aalkhuraif@ksu.edu.sa (A.A.A.-K.); 7Department of Restorative Dental Sciences, Division of Operative Dentistry, College of Dentistry, Jazan University, Jazan 44512, Saudi Arabia; shilpa.bhandi@gmail.com; 8Department of Oral and Maxillo Facial Sciences, Sapienza University of Rome, 00161 Rome, Italy; ale.zanza@gmail.com (A.Z.); luca.testarelli@uniroma1.it (L.T.); 9Department of Maxillofacial Surgery and Diagnostic Sciences, Division of Oral Pathology, College of Dentistry, Jazan University, Jazan 44512, Saudi Arabia

**Keywords:** anxiety, dental students, depression, psychological distress, Saudi Arabia, stress

## Abstract

The objective of this paper was to evaluate the studies that have reported on psychological issues among dental students in Saudi Arabia and to develop coping strategies to overcome these mental health-related issues. The present systematic review is in accordance with the guidelines for Preferred Reporting Items for Systematic Reviews and Meta-Analyses (PRISMA). The search for the articles was carried out in the electronic databases by four independent researchers. The data search was performed in the electronic search engines like PubMed, Google Scholar, Web of Science, Scopus, Medline, Embase, Cochrane and Saudi Digital Library for scientific research articles published from January 2000 until December 2020. STROBE guidelines were adopted for qualitative analysis of six articles which met the eligibility criteria. The analysis of the literature revealed that most of the studies included were conducted in the past 8 years in different regions of Saudi Arabia. Findings of this systematic review clearly state that dental students in Saudi Arabia experience higher levels of depression, stress and anxiety and stress during their education period, with a higher stress for female students compared to male students. There is an urgent need to introduce interventional programs and preventive strategies to overcome the long-term effects.

## 1. Introduction

Healthcare providers are constantly subjected to higher levels of stress which becomes an inherent part of their profession as high expectations are placed on them by their society. Dentistry is considered one of the most stressful professions and stressful experiences begin right from the commencement of their dental education. Stressful experiences eventually affect the mental and physical health of an individual which, in turn, affects their personal and professional life leading to disruption of their well-being in comparison with other health professionals [1,2]. Globally, several studies have reported that dental education is more stressful in comparison with medical and other health professions and have raised concerns [3,4,5].

Dental education is considered most stressful and was significantly found to intensify among undergraduate students as they gradually move on to successive years of study [6]. Long-term effects of stress conditions have been linked with the development of other disorders. An association has been found between higher stress levels and mental health where the stress levels have been the root cause for significant problems such as depression and anxiety [7]. Stress is considered as a psychological and physical reaction of one’s body in response to changing conditions which could be either real or perceived, positive or negative [8]. Symptoms of stress mainly include fear, fatigue, tension, instability and sleeplessness [9,10,11]. During the period of dental education, dental students undergo tremendous stress for various reasons which include demanding curriculums, meeting course and clinical requirements, handling anxious patients, limited free time, examinations, higher expectations and profound expectations with academic performance [10,12,13,14]. The demand for excellence in academic performance and other requirements in dentistry makes the preclinical to clinical transition for students even more stressful [15,16].

Stress among dental students can have a detrimental effect, significantly impact on their physical and mental health and threaten an individual’s well-being [17]. Higher levels of stress among the dental students eventually result in psychological morbidity and emotional exhaustion, which would result in professional burnout and decreased productivity [18].

Considering the higher prevalence of psychological distress among dental students in Saudi Arabia, there is a need for developing coping strategies that could be in the form of support programs and implementing preventive measures which can be beneficial to the students. The objective of this paper was to evaluate the studies that have reported on the psychological issues among dental students in Saudi Arabia and to develop coping strategies to overcome these mental health-related issues. Our paper also intends to highlight the need for policymakers and curriculum developing committees to consider these facts when developing curricula for dental education.

## 2. Materials and Methods

### 2.1. Search Strategy

The present systematic review is in accordance with the guidelines for Preferred Reporting Items for Systematic Reviews and Meta-Analyses (PRISMA) [19]. The search for the articles was carried out in the electronic databases by four independent researchers. All the electronic databases were thoroughly searched for scientific research articles published from January 2000 until December 2020. The data search was performed in the electronic search engines like PubMed, Google Scholar, Web of Science, Scopus, Medline, Embase, Cochrane and Saudi Digital Library. Combinations of Index words including “dental students”, “psychological distress”, “stress”, “anxiety”, “depression” and “mental health” were used for searching the articles. A manual search for the research articles was also conducted simultaneously.

During the preliminary search, articles were retrieved based on the title and abstract. The preliminary search yielded 134 articles that were related to our topic of interest. Due to the duplication of the data, 38 articles were eliminated. Following this, 96 articles were considered for the next stage after which the following eligibility criteria were applied.

### 2.2. Eligibility Criteria

#### 2.2.1. Inclusion Criteria

In this systematic review, cross-sectional observational studies that estimated the prevalence of stress, depression and anxiety among dental students in Saudi Arabia were included. A standard assessment tool was used for assessing mental health status.

#### 2.2.2. Exclusion Criteria

Unpublished papers uploaded online, articles with only abstracts and no full text, articles that were not in the English language and review articles were excluded from the study.

### 2.3. Study Selection

Eight articles met these eligibility criteria following which the journal details were covered and distributed among the 6 panel members for critically analyzing the included studies. The panel members were instructed to assess the quality of the report by referring to the criteria-Strengthening the Reporting of Observational Studies in Epidemiology (STROBE) [20]. Following this, the authors had discrepant opinions for the inclusion of 2 articles. These articles were excluded from the systematic review after an agreement between the panel members. Finally, 6 articles that met the eligibility criteria were considered in this systematic review for qualitative analysis (Figure 1).

## 3. Results

### 3.1. Search Results

This systematic review included six research articles that were analyzed for qualitative data. The analysis of the literature revealed that most of the studies included were conducted in the past 8 years in different regions of Saudi Arabia, mainly in Riyadh [21,22,23], Makkah [24,25] and Abha [26].

### 3.2. Data Extraction and Qualitative Synthesis

All six studies included were cross-sectional studies and the sample size included in these studies varied from 58 to 425 dental students.

Table 1 shows the details of the characteristics of cross-sectional studies that were included in this systematic review. Mental health variables of the dental students were assessed using different standardized scales; Depression, Anxiety and Stress Scale (DASS-21), Dental Environmental Stress (DES) survey, Perceived Stress Scale (PSS), Satisfaction with Life Scale (SWLS) and General Self-Efficacy scale (GSE).

The results of the included studies show that the students experienced a higher amount of depression, anxiety and stress during their period of dental education. Three studies reported that female students experienced much higher levels of depression, anxiety and stress when compared to male students [21,22,23]. One study reported that male students experienced higher levels of distress when compared to female students [25]. Another study reported similar levels of dental environmental stress among both male and female students [26].

### 3.3. Quality Assessment of the Cross-Sectional Studies Included in This Systematic Review

We analyzed the quality of the selected papers by using STROBE (Strengthening The Reporting of Observational Studies in Epidemiology) which assessed 22 key items that should be present in the title, abstract, introduction, methodology, results and discussion of a cross-sectional survey, case-control and cohort observational studies [20,27]. From this analysis, we obtained the STROBE results for each study (Table 1) as follows: 0–7 items were considered as low quality, 8–14 items were categorized as being of intermediate quality and those that included 15–22 items were considered to be high-quality articles. All the studies included in this article belong to the high-quality category.

## 4. Discussion

The dental profession is considered as one of the most stressful health professions as dental students undergo higher levels of mental health issues. It is highly demanding and dental students are bound to have stressful learning environments [28]. Dental education is considered a stressful professional course for various reasons, including an indispensable pressure to perform well academically, competing with fellow students for grades, anxiety due to exams, meeting academic and clinical requirements, worrying about career choices and limited time for relaxation [12,29,30].

In this systematic review, we intended to critically evaluate the studies that have reported on the mental health status of dental students in Saudi Arabia. Most of these studies have used standard assessment tools like the DASS-21, DES, PSS, SWLS and GSE.

The DASS-21 measures the three dimensions of these psychological conditions in a single, concise and comprehensive scale. DES questionnaire contains 41 stress-related items. PSS is a 10-item scale for assessing perceived stress levels. SWLS is a five-question scale that measures satisfaction with life. GSE is a 10-question scale that measures self-efficacy within every student.

In this systematic review, we found that most of the studies conducted in Saudi Arabia reported higher levels of stress among dental students compared to other health professional students. These results are in comparison with studies conducted among dental students in different parts of the world which also reported that dental students experience tremendous levels of depression, stress and anxiety during their dental education [31,32,33,34,35,36,37]. Competition for grades, frequent exams, less time for relaxation, deadlines for completion of tasks and loads of preclinical and clinical requirements are some of the most common stressful reasons or determinants of stress among dental students that have been reported in the literature.

There are several studies that have reported on the mental health status of dental students across the globe. In this systematic review, three out of the six included studies reported that female students in Saudi Arabia experienced a higher level of stress when compared to male students. These findings were similar to the results of other studies that have been conducted in various other countries [6,12,23,33,34,38]. This could be because female students are better expressive with their emotions or these variations can also be attributed to the fact that males usually avoid expressing or revealing their feelings when compared to females [15,16,39,40]. This could also be attributed to the fear of lack of job opportunities post education as there is some degree of gender discrimination reported. Being a health care provider is not always welcomed by the society due to the fact of being in contact with male patients including colleagues and staff, cultural restrictions, gender sensitivity and taboo which could eventually result in lower level of satisfaction among the females [41,42,43].

Previous reports suggested that dental education is more stressful than medical education. This difference has been attributed to the additional and fine psychomotor skills that are expected from the dental students in dental education [4,44].

Senior grade students consider themselves more stressed when compared with first and second grade students. There have been several reasons behind this, the most important of which include the need for finding suitable clinical patients, fear of dealing with patients, worrying about the future and also about job opportunities [12,22,28]. On the contrary, few studies have also reported that depression, anxiety and stress is seen more among first and second grade students [37,45]. This may be due to exposure to new learning environments, adaptation difficulty to the professional curriculums and course requirements, difficulty in time management and being inexperienced and less skilled in dealing with challenges.

There is a greater interest in developing management strategies to overcome the factors which increase an individual’s chances of developing psychological distress during the period of dental education. Most of the studies that have reported on the perception of depression, anxiety and stress have expressed the urgent need for addressing this issue immediately.

Hence, in this article, we intend to recommend few coping/management strategies for dental students or studies that can help them overcome the stressful phase more smoothly and have a better quality of life.

### 4.1. Coping/Management Strategies for Dental Students 

#### 4.1.1. Identify the Individuals Affected

−Students suffering from stress and depression must be identified.−Standardized assessment tools must be used for identifying individuals.−Standardized tools like the DASS-21, DES, PSS, SWLS and GSE must be used. Visual Analog Scale (VAS) that has good validity and reliability [46]. Spielberger State-Trait Anxiety Inventory (STAI) which is a 40-item Likert-type questionnaire designed to assess individual differences in experiencing anxiety and also demonstrating strong validity and reliability [47] and COPE which is a 60-item Likert-type questionnaire that evaluates an individual’s general coping methods in response to stressors 46 must be used for assessing the mental status of the individuals (Figure 2).

#### 4.1.2. Identify the Source of Stressors

−A stressor is anything that initiates or activates a reaction to stress. They are broadly categorized as academic and non-academic stressors [48].−Academic stressors include stress related to excessive workload, attendance, assessments, curriculum, competition with the grades, academic performance, clinical requirements, fear of failure and limited time for relaxation.−Non-academic stressors include trust issues, lack of self-confidence, financial stressors, unhealthy eating habits and fear of insufficient job opportunities in the future.

#### 4.1.3. Planning of Stress Management Strategies

−Strategies must be developed to reduce the fear of failure among the students along with reducing the workload pressure.−Reducing the dental curricula would be unrealistic but the curriculum development committees should consider all things that increase stress when planning the curriculum [49].−The clinical requirements should be lowered and must not be a prerequisite for passing [50].−Faculty members should schedule regular information sessions.−Faculty members should be more approachable which can improve the interpersonal relationship between the faculty and the student thereby having a positive impact on the students [51,52].−Faculty members should provide feedback to their students regularly and should work along with them for figuring out beneficial methods that could improve the student’s performance.−Faculty members should encourage and appreciate the students on their performance.−Students should be engaged in social activities which can help them develop good team spirit.−Stress management and wellness courses should be implemented in the curricula.−Study skills and time management should be emphasized and addressed regularly.−Students should be encouraged to adopt a healthy lifestyle (healthy eating, regular exercises, aerobics, adequate sleep, yoga and meditation) which promises better mental and physical health [53,54].−Students should be encouraged to identify their goals, set achievable goals and prioritize them.−Students should be encouraged to work on one task at a time. They must be always encouraged to acknowledge their stress rather than ignoring it. Students should also be trained with various efficient and practical ways of dealing with stress.−Students should be encouraged to avoid stress building/negative thoughts. Instead, they should be taught to replace them with positive thoughts or stress-buster activities.−Students should be encouraged to reach out to their family and friends frequently to develop a strong support system.

## 5. Conclusions

Dental students tend to experience tremendous stress during their education. Studies included in this systematic review clearly state that dental students in Saudi Arabia, females in particular, experience higher levels of depression, stress and anxiety in comparison to other health professional students. It is extremely essential to value the findings of these studies as they emphasize an urgent need to introduce interventional programs and preventive strategies to overcome the long-term effects of disrupted mental health or avoid mental health-related issues in the first place in young and budding dentists. It is of utmost importance for dentists to be of good mental health status, not only for their well-being, but also in the best interest of their patients.

## Figures and Tables

**Figure 1 healthcare-09-00429-f001:**
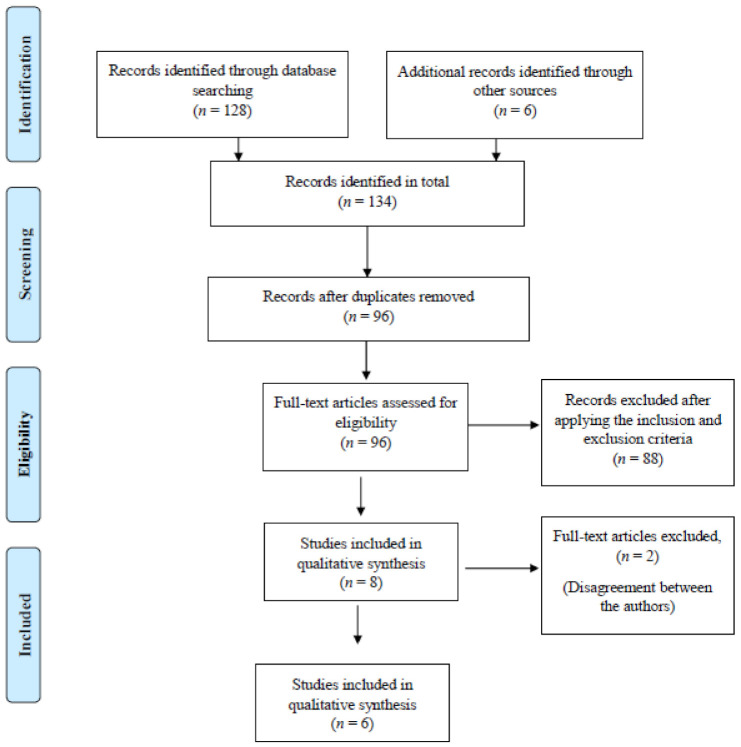
Flowchart for screening and selection of articles.

**Figure 2 healthcare-09-00429-f002:**
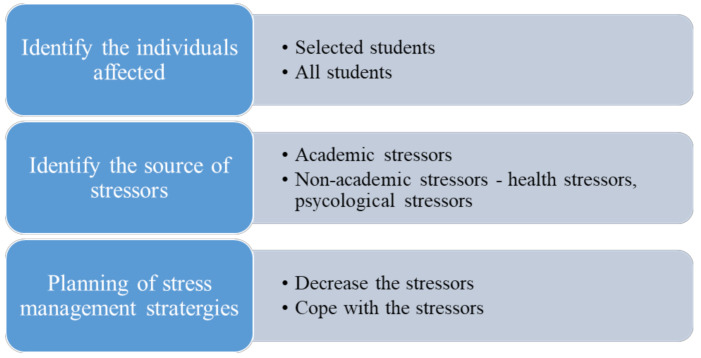
Coping/management strategies for dental students.

**Table 1 healthcare-09-00429-t001:** The characteristics of studies that were included in this systematic review.

Sl.No	Author Name and Year	Region	Study Design	Participants	Gender	Comparison if Any	Assessment Tool	Mental Health Variables	Results	Conclusions or Suggestions	STROBE
1	Al-Sowygh.et al. 2013 [21]	Riyadh	Cross sectional study	425 Dental students	Maleߞ293 Femaleߞ132	No comparison	Questionnaire-Survey	Modified Dental Environmental Stress (DES) scale		Female students perceived significantly more stress than males	High quality
2	Al-Sowygh et al. 2013 [22]	Riyadh	Cross sectional study	425 Dental students	Maleߞ293 Femaleߞ132	No comparison	Questionnaire-Survey	Dental Environmental Stress (DES) survey, Perceived Stress Scale (PSS) and Brief Coping Scale (BCS).	Scores-Mean (SD) DES scores- Malesߞ97.2(20.7), Femalesߞ104.8(22.2) PSS-Malesߞ22.02 (3.74) Femalesߞ24.59 (3.99)	High occurrence of stress scores was seen in dental students Female, advanced and married, compared with male, junior and single students reported more stress	High quality
3	Basudan et al. 2017 [23]	Riyadh	Cross sectional study	227 Dental students	Maleߞ134 Femaleߞ113	No comparison	Questionnaire-Survey	DASS-21 scale	Depressionߞ55.9%, Anxietyߞ66.8% Stressߞ54.7% Extremely severe scores for depressionߞ20.2%, anxietyߞ34.0%, and stressߞ20.2%	High occurrence of depression, anxiety and stress among dental students Females displayed higher stress and anxiety scores	High quality
4	Aboalshamat et al. 2014 [24]	Makkah	Cross sectional study	58 Dental students 259-Medical students	Not mentioned	No comparison	Questionnaire-Survey	DASS-21 scale, Satisfaction with Life Scale (SWLS), General Self-Efficacy scale (GSE)	Scores-Mean (SD) Depression-Term 1ߞ15.93 (9.23) Depression-Term 2ߞ13.1 (8.85) Anxiety-Term 1ߞ12.03 (8.23) Anxiety-Term 2ߞ10.66 (8.62) Stress-Term 1ߞ20.41 (8.66) Stress-Term 2ߞ18.41 (8.3) SWLS-Term 1ߞ24.16 (5.46) SWLS-Term 2ߞ25.52 (5.25) GSE-Term 1ߞ27.12 (4.53) GSE-Term 2ߞ27.41 (3.99)	Compared to the middle of the 1semester (T1), Depression, anxiety, and stress measured at the beginning of 2nd semester (T2) were found significantly lower, while Satisfaction with life was significantly higher.	High quality
5	Aboalshamat et al. 2015 [25]	Makkah	Cross sectional study	72 Dental students 350- Medical students	Not mentioned	Medical students	Questionnaire-Survey	DASS-21 scale, Satisfaction with Life Scale (SWLS), General Self-Efficacy scale (GSE)	Depressionߞ(69.9%), Anxietyߞ(66.4%), Stressߞ(70.9)% Scores-Mean (SD) Depressionߞ15.05 (SD¼9.12), Anxietyߞ11.98 Stressߞ(SD¼8.82), SWLSߞ23.60 (SD¼6.37) GSEߞ27.22 (SD¼4.85)	High levels of psychological distress were found. Male dental students had higher distress than female. Dental students were more depressed in the third year, but more stressed in the second year.	High quality
6	Aseeri et al. 2018 [26]	Abha	Cross sectional study	301 Dental students	Maleߞ151 Femaleߞ150	No comparison	Questionnaire-Survey	Dental Environmental Stress (DES) survey		Male and female students exhibited similar mean DES scores	High quality

## Data Availability

Data is contained within the article.

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
