# Peer review of "Psychological Distress among Undergraduate Dental Students in Saudi Arabia and Its Coping Strategies—A Systematic Review"

_healthcare, 2021, doi:10.3390/healthcare9040429_

Round 1

Reviewer 1 Report

The manuscript is generally well-written, and serves a useful (and understudied) population.

One major concern is with the gender findings realted to stress; namely, that females reported more stress than males. First, the author's contention that this may be due to females' higher neuroticism is not supportable, as such results were from US samples, and their results are not of a US sample. Second, the authors overlook a significant possible cause: the lack of support of females in a male-dominated field. There is ample evidence of a similar phenomenon in other fields (e.g., STEM), and the parallels should be explored.

Author Response

Please see the attachment. All comments have been reported there.

Reviewer 2 Report

Exact repetition of ideas, lines 59-62

Studying dentistry can be stressful, but no more than being a university professor, for example. The present research is not justified enough beyond repeating that studying dentistry produces stress and citing literature defending this point. It would be necessary to provide some comparative data.

The introduction does not go in-depth on the subject and does not justify the present case study.

Some of the hypotheses or statements raised are not scientific but political or institutional. An investigation needs to study  concrete facts and it is in the conclusions where its usefulness (or unusefulness) in society  must be indicated.

In how many languages ​​was the keyword search done? One would expect at least two, Arabic and English.
Excluding studies for not finding the full text in a first search greatly biases the study. Among one hundred studies, which already seem  not too many, only eight of them meet the criteria?

The method does not take into account other factors such as the number of references, their updating, or whether these articles are paid for by institutions or are published in journals with an impact factor (JCR, for example).

Some parts present a bias approach. For example, there are numerous studies that show that women suffer more stress than men in university studies for different objective reasons, including dentistry, citing only those saying the opposite   is not objective and reduces the scientific rigor of the present study.

The difference in stress according to gender should be studied much much more deeply.

Author Response

(The authors gave the same response as above.)

Round 2

Reviewer 1 Report

Better!